# Anisotropy of the ΔE Effect in Ni-Based Magnetoelectric Cantilevers: A Finite Element Method Analysis

**DOI:** 10.3390/s22134958

**Published:** 2022-06-30

**Authors:** Bernd Hähnlein, Neha Sagar, Hauke Honig, Stefan Krischok, Katja Tonisch

**Affiliations:** 1Technical Physics 1 Group, Institute of Micro- and Nanotechnologies (IMN MacroNano^®^), Technische Universität Ilmenau, Postfach 100565, 98684 Ilmenau, Germany; neha.sagar@tu-ilmenau.de (N.S.); stefan.krischok@tu-ilmenau.de (S.K.); 2Materials for Electronics and Electrical Engineering Group, Institute of Micro- and Nanotechnologies (IMN MacroNano^®^), Technische Universität Ilmenau, Postfach 100565, 98684 Ilmenau, Germany; hauke-lars.honig@tu-ilmenau.de

**Keywords:** delta E effect, magnetoelectric sensor, Nickel, anisotropy

## Abstract

In recent investigations of magnetoelectric sensors based on microelectromechanical cantilevers made of TiN/AlN/Ni, a complex eigenfrequency behavior arising from the anisotropic ΔE effect was demonstrated. Within this work, a FEM simulation model based on this material system is presented to allow an investigation of the vibrational properties of cantilever-based sensors derived from magnetocrystalline anisotropy while avoiding other anisotropic contributions. Using the magnetocrystalline ΔE effect, a magnetic hardening of Nickel is demonstrated for the (110) as well as the (111) orientation. The sensitivity is extracted from the field-dependent eigenfrequency curves. It is found, that the transitions of the individual magnetic domain states in the magnetization process are the dominant influencing factor on the sensitivity for all crystal orientations. It is shown, that Nickel layers in the sensor aligned along the medium or hard axis yield a higher sensitivity than layers along the easy axis. The peak sensitivity was determined to 41.3 T^−1^ for (110) in-plane-oriented Nickel at a magnetic bias flux of 1.78 mT. The results achieved by FEM simulations are compared to the results calculated by the Euler–Bernoulli theory.

## 1. Introduction

Magnetic field sensors based on electromechanical systems have gained a lot of attraction in the last decade as the magnetoelectrical sensor concept exhibits promising device characteristics enabling the detection of the weakest magnetic fields as needed for example in biomedical applications or geophysical explorations. In this kind of sensors, the ΔE effect, which describes the change of Young’s modulus in presence of a magnetic field, is the basic physical property utilized in this sensor concept. High sensitivities and low limits of detection in the low $pT/\sqrt{Hz}$ regime [1,2,3] have been realized in the last decade, paving the way down to the $fT/\sqrt{Hz}$ range at room temperature [4], where usually only superconducting quantum interference devices (SQUID) [5] could be used. Magnetoelectric sensors exhibit the great advantage that in contrast to SQUIDs no extensive cooling is required to achieve their superconducting state for operation, leading to less complex and costly operation. Currently, common magnetoelectric sensors exhibit sizes in the millimeter [3,6,7] up to the centimeter range [8,9,10] and are usually based on amorphous soft magnetic materials, for example FeCoSiB [11], FeGaB [12] or Terfenol-D [2], in combination with a piezoelectric material for the output signal generation, such as polycrystalline AlN [12], single crystalline PZT [13] or PMN-PT [14]. However, hard magnetic materials such as Nickel are also in the focus of research. In magnetoelectric sensors, Nickel is often used in combination with a second magnetostrictive material to realize magnetization-graded structures for the optimization of the magnetoelectric coupling [15,16]. Nickel is also an easy-to-process material for thin-film applications and, compared to magnetostrictive compound materials, relatively simple to describe as a model system. Additionally, if necessary, it provides the possibility to be processed as a single crystalline material, e.g., an inverted stack allows the deposition of AlN on (polycrystalline) Ni foils, which can be replaced with single crystalline ones, with a high interface quality and c-axis orientation of AlN, resulting in a strong magnetoelectric response compared to Fe and Co [17]. Comparably few references can be found that target sensors in the microelectromechanical system (MEMS) regime [18,19,20], a highly interesting transition region where anisotropic material properties show increasing influence on the device characteristics, when the sensor dimensions start to reach the order of magnitude of individual crystals within the (poly-)crystalline material. However, in small MEMS structures size effects also start to play a role, e.g., influencing Young’s modulus [21] and the magnetization [22] of Nickel or the piezoelectric layer [23,24]. This makes a detailed investigation of the material and scaling properties inevitable.

In recent studies, the properties of MEMS structures were investigated in regards to the implications of anisotropy of the elastic [25] and the magnetoelastic parameters [26]. The angle-dependent analysis of the ΔE effect sensors based on TiN/AlN/Nickel revealed a complex eigenfrequency behavior in the presence of a magnetic field (see Figure 1a,b). It was found that uniaxial magnetic anisotropy is able to describe the ΔE effect in general and that additional anisotropy contributions besides the shape anisotropy are too complex for fitting the results within an analytic model. To identify different contributions from magnetic anisotropy, finite element simulations are nowadays a powerful tool based on advanced and well-tested models.

To investigate the intrinsic potential of tuning the crystalline texture of the Nickel as a magnetostrictive material, the anisotropic properties and the respective dependency of the sensitivity on different crystal orientations are analyzed within the proposed finite element study. Another design parameter to be optimized depending on the cubic anisotropy is the Nickel layer thickness. The layers are modeled as single crystals to be able to study the different effects from the point of view of a most general (ideal) case. This allows the study of the intrinsic anisotropic behavior of Nickel while minimizing the influence of the specific lateral sensor design which has usually a high impact on the device performance.

## 2. Model Details

### 2.1. Analytic Description of the ΔE Effect in Nickel

Magnetostriction in general describes the structural response of the lattice of a ferromagnetic material to the change of an external magnetic field. The magnetic domains in ferromagnetic materials such as Nickel are randomly oriented in the unmagnetized state, each with saturation magnetization $M_{s}$. In the presence of a magnetic field, the minimization of the internal energy density function *u*, Equation (Equation 1), leads to an increasingly parallel alignment of the domains along the field direction until saturation.
(1)$$u=u_{Z}+u_{sh}+u_{\sigma }+u_{cr}+\ldots$$

The energy density function contains contributions from the Zeeman energy $u_{Z}$ describing the dependency of the external field strength *H* and the magnetization direction as well as several anisotropy terms. Those are the shape anisotropy $u_{sh}$, which considers demagnetizing effects, the magnetoelastic anisotropy $u_{\sigma }$, taking into account mechanical stresses, and the crystalline anisotropy $u_{cr}$. In this work, only the latter for the case of a cubic crystalline anisotropy and the Zeeman energy were considered. The influence of the remaining contributions is discussed in Section 2.3 [26].

In isotropic or ideal polycrystalline materials the magnetostrictive strain λ in a direction θ relative to the magnetization direction is given by [27]
(2)$$\lambda =\frac{3}{2}\lambda_{s}\left(cos^{2}\theta -\frac{1}{3}\right),$$
with $\lambda_{s}$ as the isotropic saturation magnetostrictive strain. In the anisotropic case, the magnetostriction is dependent on the principal axes (hkl) of the materials lattice. The magnetostriction is thus given (and used by Comsol) by
(3)$$\lambda_{hkl}=\frac{3}{2}\lambda_{100}\left(\sum_{i=1}^{3}m_{i}^{2}\psi_{i}^{2}-\frac{1}{3}\right)+3\lambda_{111}\left(\sum_{i,j=1}^{3}\vec{m_{i}}\vec{m_{j}}\psi_{i}\psi_{j}\right),$$
with i≠j and cyclic permutation. Here, $\psi_{i,j}$ is the angle cosine of the respective direction in relation to $\vec{M}$ and $\vec{m_{i,j}}$ the direction vector of $\vec{M}/M_{s}$. Within Equation (Equation 3), the volume conservation is assumed. Other effects breaking the volume conservation such as the volume magnetostriction [28] were thus not considered. The magnetostriction constants of Nickel are all negative [29] leading to compression strains along the three principal axes. The experimental curves are given in Appendix A with their respective fits. The curve for the (110) direction is also presented for completeness, though not needed according to Equation (Equation 3). As the curve fits exhibit an increasing error for H< 1000 A/m, the discussion of the sensor characteristics in Section 3 is limited to B> 1 mT.

The development of magnetically induced strains $\lambda_{hkl}$ in a magnetostrictive material results in a change of Young’s modulus, the so-called ΔE effect. It can be described analytically [30], so that
(4)$$\frac{1}{E_{hkl}^{Ni}}=\frac{\partial (\epsilon_{hkl}+\lambda_{hkl})}{\partial \sigma_{hkl}}=\frac{1}{E_{hkl,sat}^{Ni}}+\frac{1}{\Delta E_{hkl}^{Ni}},$$
while $\Delta E_{hkl}$ is directly dependent on the derivations of $\lambda_{hkl}$ and the magnetization $M_{hkl}$:(5)$$\frac{1}{\Delta E_{hkl}^{Ni}}=\frac{{\left(\partial \lambda_{hkl}/\partial H\right)}^{2}}{\mu_{0}\partial M_{hkl}/\partial H}.$$

The static inverse elastic modulus $1/E_{hkl,sat}$ of the cubic Nickel lattice in Equation (Equation 4) can be calculated using the compliance matrix $S_{ii}$ and the direction cosines α, β and γ by [31]:(6)$$\frac{1}{E_{hkl,sat}^{Ni}}=S_{11}-\left(2\left(S_{11}-S_{12}\right)-S_{44}\right)\left(\alpha^{2}\beta^{2}+\alpha^{2}\gamma^{2}+\beta^{2}\gamma^{2}\right).$$

The tensor elements of the compliance matrix for Nickel were derived from the literature values [32] and averaged to $S_{11}=7.47\times 10^{-12}{\mathrm{Pa}}^{-1}$, $S_{12}=-2.84\times 10^{-12}{\mathrm{Pa}}^{-1}$ and $S_{44}=8.33\times 10^{-12}{\mathrm{Pa}}^{-1}$ (see Appendix A). The resulting Young moduli for the three directions in saturation were $E_{100,sat}=$ 134 GPa, $E_{110,sat}=$ 228 GPa and $E_{111,sat}=$ 297 GPa, respectively. The Poisson ratios for the single crystalline Nickel derived from the elastic constants equaled $\nu_{\left(\langle 1,0,0\rangle ),(\langle 0,1,0\rangle \right)}=0.381$, $\nu_{\left(\langle 1,1,0\rangle \right),\left(\langle 1,\bar{1},0\rangle \right)}=-0.06$ and $\nu_{\left(\langle 1,1,1\rangle \right),\left(\langle 1,\bar{1},0\rangle \right)}=0.142$. $\nu_{\left(\langle 1,0,0\rangle ),(\langle 0,1,0\rangle \right)}$ was thus higher than reported values for other single crystals (0.315–0.329) [33] or nanowires (0.305–0.335) [34]. A stiffness matrix could then be generated based on the nonlinear $E_{hkl}^{Ni}$ curves (similar to TiN discussed later in this section), which was independent from stress anisotropy.

In the cubic crystal, different transitions are passed through during magnetization. For low fields, domain wall shifts appear which are followed by the domain rotation out of the easy axis towards the external magnetic field direction, eventually followed by the magnetization reversal of antiparallel-oriented domains and saturation. In Nickel, the (111) direction is the easy axis and the (110) and (100) directions are the medium and hard axes, respectively. This is depicted in Figure 2a in combination with the derived Young moduli $E_{hkl}$ according to Equation (Equation 4).

The domain wall shift region is beyond the accurate fit limit, which is why an investigation in this range is not possible with the given data. The domain rotation region describes the reorientation of the domains along the hard axis and exists only for the hard axes of magnetization. The transition from the wall shift to the domain rotation introduces a decrease in $E_{100/110}$ with a distinct minimum at around 2 mT. At higher fluxes, $E_{100/110}$ increases again and reaches a maximum in the saturation region. The easy axis is in contrast characterized by the direct transition from domain wall shifts to magnetic domain reversal, as the magnetic domains are already aligned along (111). This leads to a rather flat dependency up to the point where saturation happens at H> 6000 A/m. This is accompanied by a strong increase in Young’s modulus, the known effect of magnetic hardening. This is also observable for the (110) direction, while it has to be noted that the (100) direction exhibits no such hardening, i.e., $E_{100,0}=E_{100,sat}$. Not only is the shape of the ΔE effect direction-dependent but also the magnitude. In the (100) direction, the maximum change of Young’s modulus $E_{100,min}/E_{100,sat}$ is 7.5%. The other two directions exhibit a much higher change of 17%, which is similar to the reported value of 20% [35]. Observations in Nickel nanocrystals revealed even increases of 31% along (111) [35], nearly twice as high as the calculated increase in this work.

### 2.2. Description of the Finite Element Model

The sensor design was based on a three-layer cantilever structure with the length $l_{c}=$ 25 μm and width $w_{c}=$ 4 μm investigated recently [26] and shown in Figure 1a,b. Accordingly, the respective layer parameters were derived from the experiment and given by TiN (90 nm) as the back electrode, AlN (450 nm) as piezoelectric and the magnetostrictive Nickel on top with varying thickness $t_{Ni}$ between 0 and 1000 nm. In contrast to the experimental structures, where the actual distribution in size and orientation of the Nickel polycrystals were unknown, all materials were treated as a single crystal within this study as it allowed us to investigate the sensor characteristics solely in regard to the respective Nickel crystal orientation. For the investigation of the vibrational behavior with Comsol 5.6 (Comsol Multiphysics GmbH, D-37073 Göttingen, Germany) a 2D model with coupled multiphysics (solid mechanics and magnetostriction) was used to minimize the calculation time while allowing a high accuracy using dense meshes. The precise parameters are given later in this section. AlN and TiN were treated as anisotropic linear elastic materials following the relationship between the strain tensor ϵ and the stress tensor σ of Hooks’s law:(7)$$\epsilon_{ij}=C_{ijkl}^{-1}\sigma_{kl},$$
with $C_{ijkl}^{-1}$ being the compliance matrix of the respective material within the stack. For TiN with the cubic fcc lattice and space group Fm$\bar{3}$m, as well as for a hexagonal AlN with space group P63mc, the $C_{ijkl}^{-1}$ are given in Appendix B. The orientation of the AlN layer is fixed with the c-axis perpendicular to the film plane, just as in the experimental structures, and a <100> direction is fixed in the cantilever length direction, contrarily to the experimental random in-plane orientation distribution. The latter simplification should be negligible as there is only a weak mechanical anisotropy within the AlN’s basal plane. For TiN the cubic [100] axis was oriented perpendicular to the film plane and [010] into the cantilever direction.

The sensor model contained two different sets of boundary conditions. Within the structural mechanics domain, fixed constraints were applied at the left end of the multilayer structure (see Figure 1c or Figure 3a). In the magnetic domain, the boundaries of the Nickel layer were set to be magnetically insulated. This led to a homogeneous distribution of the magnetic flux in the magnetostrictive material, and it suppressed the formation of shape anisotropy and design-dependent magnetic stray fields. The direction of the external magnetic flux *B* was kept constant throughout the analysis and applied along the longitudinal direction of the cantilever. *B* was logarithmically scaled between 1 mT and 400 mT using 54 sampling points.

The static mesh was adjusted to the respective layers with a quad mesh for TiN (mesh size of 106,250 elements), a triangular mesh for AlN (mesh size of 276,812 elements) and a quad mesh for Nickel (mesh size of 100,000 elements). The minimum and maximum element sizes were 1 nm/20 nm, 10 nm/100 nm and 1 nm/$\left(t_{Ni}/20\right)$ for TiN, AlN and Nickel, respectively. The element growth rate was 1.1 and constant for all layers. Accordingly, the minimum element quality was >0.5. The model was solved using a linear, fully coupled (stationary and eigenfrequency), direct MUMPS solver with tolerances ≤$10^{-6}$. The convergence curves of the magnetic potential and the displacement field are given in Appendix C. It is noticeable that the convergence rate decreased with an increasing magnetic/magnetostrictive load. The stationary solution of a cantilever with $t_{Ni}=$ 100 nm is shown in Figure 1c.

### 2.3. Limits of the Proposed Model

As different constraints were made to the model to be able to investigate the crystalline anisotropy solely, the generated results are less suited to describe the specific behavior of experimental sensors (as in Figure 1a,b), but should be seen as an approach to find an intrinsic limit of the anisotropic sensitivity. As a consequence of the proposed (2D) model, the analysis of effects arising from the complex state of experimental (polycrystalline MEMS) structures is not possible, e.g., a non-rectangular cross-section with specific surface and interface roughness of the different layers affecting the shape anisotropy or contributions of other anisotropy types such as the uniaxial anisotropy. The complex stress distribution in such structures induced by the undercut thermal treatment or the growth-induced stress anisotropy, influences the eigenfrequency, especially at shorter cantilevers [25]. Within the single crystalline approximation, mechanical or magnetic effects arising from defects (vacancies, interstitial and substitution atoms, dislocations, grain boundaries) in (textured) polycrystalline layers cannot be reproduced.

## 3. Results

The calculated natural eigenfrequencies in magnetic saturation for the cantilevers with different crystal orientations of the Nickel layer are given in Figure 2b. In addition, analytically derived curves from the generated ΔE effect curves using the Euler–Bernoulli theory are shown for comparison. Here, the eigenfrequencies i of a cantilever are given by
(8)$$f_{i,c}=\frac{{\varkappa_{i}^{\prime }}^{2}}{2\pi {l_{c}}^{2}}\sqrt{\frac{{\left(E\left(H\right)I\left(H\right)\right)}_{tot}}{{\left(\rho A\right)}_{tot}}},$$
with $\varkappa_{i}^{\prime }$ as the curvature-dependent eigenvalue, ${\left(E\left(H\right)I\left(H\right)\right)}_{tot}$ as the bending stiffness of the multilayer stack and ${\left(\rho A\right)}_{tot}$ as the reduced mass. For low thicknesses $t_{Ni}$, the eigenfrequencies converge to the natural frequency of the residual layer stack of TiN/AlN at 1.312 MHz for the FEM calculation and at 1.35 MHz for the Euler–Bernoulli theory, which is in good agreement. In accordance to the change in $E_{hkl,sat}$ for the respective directions, the eigenfrequencies show the expected increase when rotating the crystal orientation away from (100) and reaching magnetic saturation. However, the results from the Euler–Bernoulli theory exhibit stronger deviations from the results of the FEM simulation. While the softening for the (100)-oriented Nickel can be reproduced by both, the minimum of the softening is shifted towards a higher $t_{Ni}$ within the analytic approach. Above approximately 300 nm in the case of (100)-Ni and 200 nm for the (110)/(111)-Ni, the eigenfrequencies scale linearly according to the simulation, which can be observed qualitatively also using Equation (Equation 8). It appears, that the Euler–Bernoulli theory is not able to reproduce the FEM results quantitatively for the magnetic saturation. The reason might be the lack of additional influencing factors that are not covered by the Euler–Bernoulli theory, such as the anisotropic material properties of the material stack or the Zeeman energy in Equation (Equation 1). The simulated eigenfrequencies at $t_{Ni}=$ 100 nm are 20–30% higher than the respective experimental eigenfrequencies [26] of the structures in Figure 1a, on which the model is based. The main reason is the undercut that was neglected in the simulation, which can lead to a frequency shift in the range of 20% [25] or even higher, depending on the undercut depth. A second important influencing factor is the single crystal approximation of the individual layers in contrast to the experimental data.

### 3.1. Magnetostriction and Bending

In presence of a magnetic field, the magnetostriction $\lambda_{hkl}$ in Equation (Equation 4) applies stresses to the Nickel layer and hence to the adjacent AlN resulting in a bending of the cantilever. As $\lambda_{hkl}$ is always negative in Nickel, cantilevers are bent upwards for the modeled stacking order, regardless of the crystalline orientation. The magnetostriction-induced tip deflection is presented in Figure 3a for the three orientations and different thicknesses $t_{Ni}$.

In magnetic saturation, the tip deflection is mainly reflecting the different saturation magnetostriction values (see Appendix A), where the (100) direction exhibits the largest strain and the (111) direction the lowest. The tip deflection is maximized at $t_{Ni}=$ 400 nm for (100) and at 300 nm for the (110)/(111) orientation and thus at the same $t_{Ni}$ where the linear eigenfrequency region in Figure 2b begins. A further increase of $t_{Ni}$ leads to a decrease of the deflection which is caused by the shift of the neutral axis towards the Nickel layer within the cantilever, leading to a decrease of the bending moment. The absolute deflection is in the range of 4–6.5 nm, which is small compared to the total thickness of the cantilever of around 1 μm. The curvature of a cantilever has influence on its eigenfrequency behavior. This consideration is important for parameter extraction, where the Euler–Bernoulli theory is used [25]. Using Equation (Equation 8), the bending induced cantilever deflection allows an estimation of the shift of its eigenfrequency in magnetic saturation. Here, $\varkappa_{i}^{\prime }$ was calculated by [36]
(9)$${\varkappa_{i}^{\prime }}^{2}=\sqrt{{\varkappa_{i}}^{4}+c{\vartheta_{i}}^{4}},$$
with $c={\left(l_{c}/r\right)}^{2}$ as the curvature coefficient, where *r* is the curvature radius and $\vartheta_{i}$ is the curvature correction term given by $\vartheta_{i}^{4}=\left(p_{1}+p_{2}k\right)/\left(p_{3}+p_{4}k\right)$, where $k=Al_{c}^{2}/I$. For the analysis of the impact of the cantilever deflection according to Figure 3a on its natural mode (i=1) in Figure 2b, the respective parameters were set to $\varkappa_{1}=1.875$, $p_{1}=0.7365$, $p_{2}=-0.5017$, $p_{3}=1.215$ and $p_{4}=-1$. The curvature-dependent change in the eigenfrequency $f_{i,c}/f_{i,0}$ is presented in Figure 3b. The respective parameters in Equation (Equation 8) were taken from the model parameters in saturation. Interestingly, the relative impact on the eigenfrequency is not directly dependent on the Nickel crystal orientation, but only on the curvature radius. Assuming a circular arc, the deflection was coupled with the curvature radius by $\delta =r(1-cos\left(l_{c}/r\right))$. This dependency is also plotted in Figure 3b along with the maximum deflections gathered from Figure 3a. The curvature resulting from magnetostriction alone within the proposed model was very small and the curvature radius comparably large. Consequently, the relative change in eigenfrequency was also very small for any crystal orientation, while for (100), the highest, and for (111) the lowest deviation can be observed. The difference between the directions was approximately 24%, having almost no effect at such low radii. However, if a prestressed cantilever is given with a static radius much smaller than 0.1 m as shown in thin AlN layers [23], the anisotropic effect of the crystal orientation should be considered in the bending correction of the eigenfrequency in Equation (Equation 8).

### 3.2. Eigenfrequency Behavior in the Magnetic Field

For an easier comparison of the different anisotropic eigenfrequency curves, their relative change $f\left(B\right)/f_{sat}$ was chosen. These are shown for different $t_{Ni}$ in Figure 4, as well as the respective specific sensitivities ∂f/∂B for each crystal orientation. Independent on the orientation, the magnitude of the ΔE effect increases with increasing $t_{Ni}$ until a steady state is reached. While the ΔE effect magnitude for (110)- and (111)-Ni starts to saturate at approximately 200 nm, in the (100) orientation, saturation appears noticeably between 300 and 400 nm. The individual saturation thicknesses of the ΔE magnitude are of particular interest in the sensor fabrication. Due to the adjacent functional layers in the sensor structure, the magnitude is decreased compared to the pristine ΔE curves of Figure 2a. In the (100) orientation the magnitude is reduced from 7.5% to 2.1% while for the (110)- and (111)-oriented Nickel it is reduced from 17% to 5.4% and 4%, respectively. This decrease in the magnitude also affects the sensitivity and cannot be avoided in a magnetoelectric MEMS device, but only minimized by design optimization. For comparison, $f\left(B\right)/f_{sat}$ generated by the Euler–Bernoulli theory is presented in Appendix D. As the dependencies are qualitatively the same as in Figure 4, these results are interpreted later in terms of sensitivity. The specific sensitivity ∂f/∂B shows a similar dependency between the hard axes with an identical absolute sensitivity maximum at 1.78 mT. This maximum is related to the transition from domain shift to domain rotation and thus is not observable for the easy axis. The local maximum of (111)-oriented Nickel at approximately 1.5 mT is originating from a steeper slope of the magnetostriction at this point and strongly dependent on the fitting accuracy in this approach. In contrast to the visual appearance of the $f\left(B\right)/f_{sat}$ curves, ∂f/∂B is absolutely much higher in the transition from domain shift to rotation at 1–2 mT than for the transition from domain rotation to magnetization reversal at > 2 mT. The difference in ∂f/∂B is an order of magnitude for (100) while for (110) it is a factor of approximately three. That is, ∂f/∂B is mainly driven by ${\left(\partial \lambda_{hkl}/\partial H\right)}^{2}$ for the hard axes and less dependent on the regime of magnetization reversal. For (111)-oriented Nickel, the highest ∂f/∂B can be found at the magnetization reversal transition as expected from the $f\left(B\right)/f_{sat}$ curve. Though the specific sensitivity shows high absolute values, the general characteristic of the curves is nonlinear, independent from the orientation of the Nickel. As a consequence, the dynamic range in terms of the usable bandwidth of the magnetic flux that can be used for magnetic field detection is quite small. For (100)-oriented Nickel, the dynamic range determined for ∂f/∂B≈0 at the tails of the point of highest sensitivity yields 2.9 mT. For the (110) and (111) orientations 1.8 mT and 8 mT can be found, respectively. However, these values derived from crystalline anisotropy can/will be affected by other anisotropy contributions in real sensors. Within MEMS-based sensors, a decrease of ∂f/∂B is to be expected due to the impact of stress anisotropy at the supporting region.

The sensitivity as a figure of merit of a magnetoelectric, mechanical sensor, e.g., singly clamped cantilevers or doubly clamped beams, is usually described by the normalized ∂f/∂B:(10)$$S_{H}=\frac{1}{f_{sat}}\frac{\partial f}{\partial B}.$$

In Figure 5a the peak sensitivities of the three principal axes of Nickel are presented for the FEM results as well as for the results gained from the Euler–Bernoulli theory in dependency of $t_{Ni}$. The sensitivities for the latter are based on the normalized eigenfrequency curves shown in Appendix D. Independent of the approach, the used models exhibit a saturation behavior with observable sensitivities nearly twice as high for Euler–Bernoulli-derived data compared to the FEM results. Additionally, the saturation thicknesses are shifted by a similar factor of around two to higher $t_{Ni}$ for the Euler–Bernoulli-generated curves leading to much less accurate results when using Equation (Equation 8), similar to the observations made in Figure 2b. The saturation thicknesses depend on the specific layer configuration in terms of materials chosen for the back electrode and the piezoelectric material as well as their respective thicknesses. For the given stack of 90 nm TiN and 450 nm AlN, the saturation thicknesses were calculated numerically to approximately 500 nm, 400 nm and 300 nm for (100), (110) and (111) and appear to scale antiproportionally with $E_{hkl}$. The saturation region is strongly dependent on the stresses and the internal magnetic stray field of the magnetostrictive layer. This can result in a decrease of the sensitivity at increasing thicknesses [11]. In soft magnetic materials with usually positive magnetostriction, such as the frequently used amorphous FeCoSiB [37], FeGaB [38] and Terfenol-D [39], internal strains (e.g., stress gradients arising from layer growth) lead to a decrease of the ΔE effect according to Equation (Equation 4). However, such strains can be minimized or tuned experimentally by controlling the growths conditions, e.g., via the substrate temperature or an applied substrate bias, via a DC offset applied to the piezoelectric layer [40] or by using a symmetric sensor design. A great benefit of Nickel in this case are the negative saturation magnetostriction constants for all axes. Nickel grows typically tensile-strained on AlN layers leading to a potentially increased magnetostriction [41] and thus ΔE effect. Similarly, this was used to optimize the sensor performance based on FeCoSiB [42]. In an otherwise unstressed cantilever, magnetostrictive bending has negligible influence on the eigenfrequency and thus sensitivity. The magnetic stray field as the second influencing factor affects the sensor performance when vertical domain separation occurs, which is usually negligible within the thin layers of MEMS structures. For comparison, the domain wall size in Nickel is approximately 125 nm [43] with typical domain sizes of about 200 nm in the unmagnetized state at room temperature [44]. Consequently, the saturation regime of the numerically derived sensitivities in Figure 5a should be a good/better estimation for real sensors than the sensitivities determined by the Euler–Bernoulli theory.

In Table 1, the extracted peak sensitivities are summarized in comparison with experimentally derived sensitivities of the structures in Figure 1a,b, as well as of magnetoelectric sensors based on other material combinations. A clear gap is visible in the experimentally realized sensors compared to the theoretical expectations. The recently measured sensitivity of hard magnetic polycrystalline Ni/AlN/TiN sensors lies in the order of 1 T^−1^ which is comparable to other references based on soft magnetic FeGa or FeCo compounds. Sensors based on FeCoSiB are able to reach higher sensitivities by a factor of 5–10 in combination with a high degree of optimization. The theoretical results still remain significantly higher but are similar between soft magnetic FeCoSiB and the (110)-oriented Nickel. The given simulated FeCoSiB sensitivity of 48 T^−1^ is obtained for the second bending mode, which yields a higher value than the first/natural mode. The first bending mode should yield a sensitivity approximately 20% lower according to the data in [11], leading to an almost identical result as Ni(110). The direct growth of (110) in-plane-oriented Nickel is experimentally difficult on a hexagonal substrate such as AlN. However, there are approaches using 150 nm thick Au/Ge interfacial layers [10] for larger sensors. Additional interface engineering is needed to see whether this configuration can be scaled down to MEMS structures. Further similarities between FeCoSiB and Nickel apply to the saturation magnetostriction [45] or the density [46] leading to a similar mass inertness in the vibrational behavior, e.g., in passive operation. However, MEMS structures are less suitable for passive operation due to the size dependence of the limit of detection [47]. In actively operated sensors, the limit of detection plays a negligible role, which is why the sensitivity in combination with the dynamic range are the figures of merit to be used.

In Figure 5b a correlation between the extracted dynamic range for the specific orientations and their sensitivity is presented. A decreasing dynamic range with increasing sensitivity is found. In accordance to the discussion of Figure 4, the dynamic range derived solely from crystalline anisotropy might/will not resemble the properties of real sensors as it is affected by contributions of other anisotropies. However, such an antiproportional coupling can also be observed for Hall effect sensors [49] using a modified bias current; the similarity found here is presumably by accident. On the basis of the presented results a stepwise integration of experimental conditions can be realized in further studies, for example in terms of polycrystallinity, stresses or design related changes. The knowledge gained in the MEMS regime might also help understand and optimize larger sensors.

## 4. Conclusions

In this work the anisotropic ΔE effect of Nickel was used to study its influence on the sensitivity of a magnetoelectric sensor within a finite element simulation approach based on recent experimental results and to evaluate the intrinsic potential of this hard magnetic material. For the three principal axes of the fcc lattice, the anisotropic Young modulus of single-crystalline Nickel was derived from the direction-dependent magnetostriction and magnetization and its elastic constants. It could be shown that the resulting magnetic field dependency of Young’s modulus was highly dependent on the orientation of the crystal and the different transitions between domain wall shift, rotation and the reversal of magnetization. As a result, the known magnetic hardening effect of Nickel could be reproduced field dependently for the (110) medium axis and the (111) easy axis in an in-plane orientation while the (100) hard axis did not exhibit this effect. The magnitude of the intrinsic ΔE effect of Nickel was anisotropic with peaks at 7.5% for (100) and at 17% for the (110) and (111) orientations, respectively. Within the sensor, the ΔE effect magnitude decreased to 2.1% for the (100), 5.4% for the (110) and 4% for the (111) orientation. The magnetostriction-induced bending of the cantilever was investigated to determine its impact on the eigenfrequency. It was shown that magnetostriction alone has negligible influence on the eigenfrequency of cantilevers. The impact of the different transitions in the magnetic field on the eigenfrequency and on the sensitivity showed that the transition from the domain wall shift to the domain rotation in the hard axes directions led to a strong sensitivity, especially along (111), yielding $S_{H}=\;41.3$ T^−1^ at a magnetic bias flux of 1.78 mT. Such a high sensitivity is nearly identical to that of frequently used soft magnetic materials, such as FeCoSiB. However, the comparison between simulations and experiment was limited due to some assumption of the simulation that were in the general case not true in real sensor samples and partly not easy to achieve even if they would lead to an improved sensitivity. Especially, the magnetostrictive film is normally not single crystalline and it is more realistic to generate a polycrystalline microstructure with a strong preferential orientation. Other limitations, as listed in Section 2.3, were the 2D instead of 3D modeling, which changed the shape anisotropy behavior, and also of course no defects such as point defects or dislocation were considered, which would change the magnetization behavior. In any case, the simulation results gave a good indication of a high potential for further optimizations of the sensor performance, regardless of the used material.

## Figures and Tables

**Figure 1 sensors-22-04958-f001:**
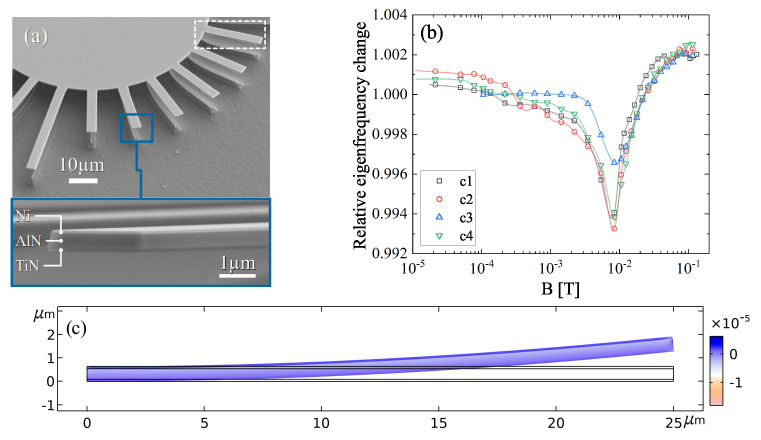
(**a**) SEM images of 4 μm wide magnetoelectric cantilevers consisting of a TiN( 90 nm)/ AlN( 450 nm)/Ni( 100 nm) layer stack investigated in recent work [26]. (**b**) Eigenfrequency characteristics in dependency of the magnetic flux of four 25 μm long and identically aligned cantilevers as marked in (**a**). (**c**) Solution of the 2D model used for the simulation study with the layer configuration from (**a**). The bending effect due to magnetostrictive strains in the 25 μm cantilever is upscaled for better visibility.

**Figure 2 sensors-22-04958-f002:**
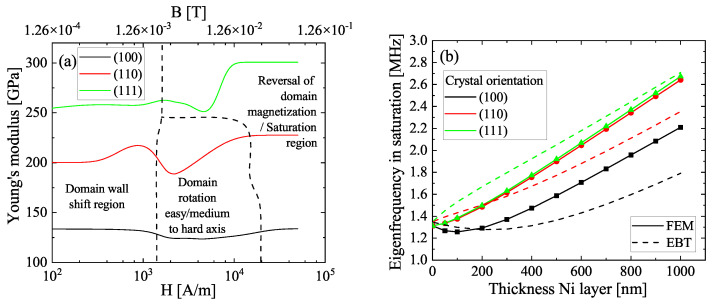
(**a**) Computed magnetic-field-dependent curves of Young’s modulus for the three principal axes. (**b**) Eigenfrequencies of the simulated cantilevers in magnetic saturation as a function of the crystalline orientation of the Nickel layer and its thickness.

**Figure 3 sensors-22-04958-f003:**
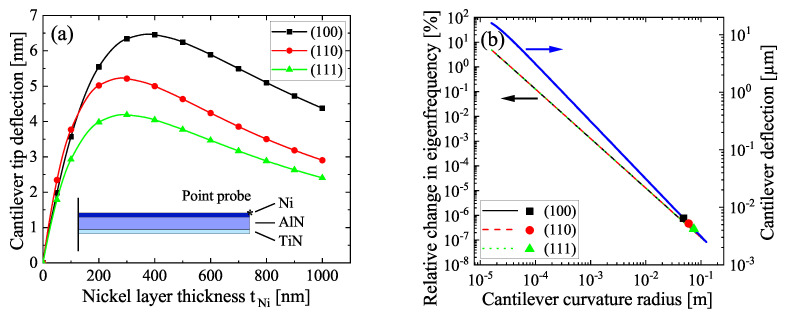
(**a**) Deflection in magnetic saturation of the simulated cantilever for different thicknesses $t_{Ni}$. (**b**) Influence of the cantilever curvature on the eigenfrequency. The curvature caused by magnetostriction is derived for comparison from (**a**).

**Figure 4 sensors-22-04958-f004:**
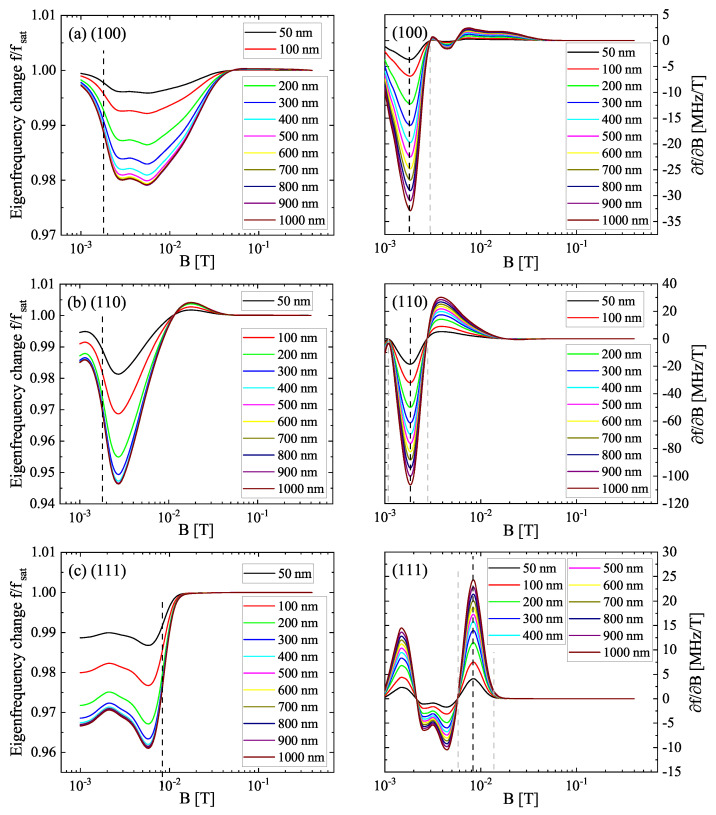
Relative eigenfrequency change $f/f_{sat}$ (left hand side) and the respective specific sensitivity ∂f/∂B (right hand side) of the three principal axes (**a**) (100), (**b**) (110) and (**c**) (111) for Nickel layer thicknesses $t_{Ni}$ in the range of 50–1000 nm. The point of highest absolute sensitivity as well as the dynamic range is marked by the lines, respectively.

**Figure 5 sensors-22-04958-f005:**
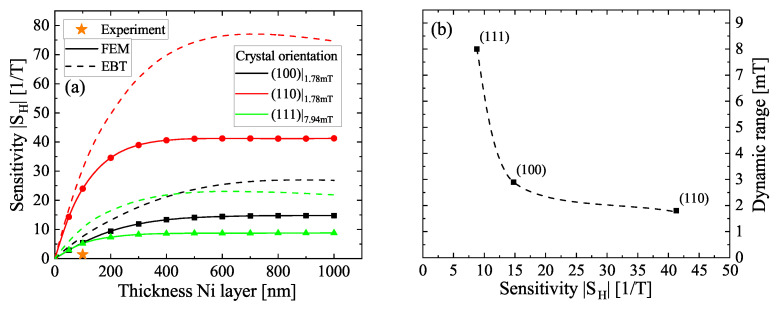
(**a**) Absolute values of the maximum sensitivities derived from Figure 4 (FEM) and from the Euler–Bernoulli theory (EBT) for the three principal axes and $t_{Ni,sat}$ at the given offsets of the magnetic flux. The experimentally achieved sensitivity is added for comparison. The value for zero thickness is extrapolated. (**b**) Extracted dynamic range in dependence of $|S_{H}|$ for the three orientations.

**Table 1 sensors-22-04958-t001:** Comparison of simulated and experimental sensitivities of the natural frequency of electromechanical system based on magnetoelectric sensors. (Values calculated according to Equation (Equation 10) if not given in the reference). * Sensitivity for the second eigenmode.

Material	Reference	Sensitivity (1/T)
Ni(100)/AlN/TiN${}^{sim}$	this work	−14.9
Ni(110)/AlN/TiN${}^{sim}$	this work	−41.3
Ni(111)/AlN/TiN${}^{sim}$	this work	8.8
poly-Ni/AlN/TiN${}^{exp}$	[26]	−0.9 …−1.4
FeCoSiB/poly-Si/AlN${}^{exp}$	[11]	10
FeCoSiB/poly-Si/AlN${}^{exp}$	[11]	13 *
FeCoSiB/poly-Si/AlN${}^{sim}$	[11]	48 *
FeCoB/Al/AlN/Pt${}^{exp}$	[19]	−0.7
FeGaB/AlN/Pt${}^{exp}$	[48]	−2.2
FeGa/Ti/Diamond${}^{exp}$	[20]	0.5

## Data Availability

Additional data are available upon request from the authors.

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
