# Peer review of "Anisotropy of the ΔE Effect in Ni-Based Magnetoelectric Cantilevers: A Finite Element Method Analysis"

_sensors, 2022, doi:10.3390/s22134958_

Round 1
Reviewer 1 Report
This manuscript only presents the results of numerical simulations of anisotropic properties of Nickel using commercial software COMSOL. However, this manuscript did include novel analytical models or experimental results to validate the simulation results.
Experimental results should be reported. The abstract and introduction are confusing.
Introduction should include more information of the main advantages and limitations of other investigations about Nickel used as magnetoelectric sensor.
What is the main scientific contribution of the proposed investigation?
The description of the sensor modelling must be significantly improved.
Which are the main assumptions of the finite element method (FEM) models?
The main load and boundary conditions, mesh type, and analysis type of the FEM models should be considered.
Experimental results should be included.
The discussions of the results shown in figures3-5 should be enhanced.
Which are the main limitations of the proposed models?
Author Response
Dear Reviewer,
please find our response attached.
Best regards.

Reviewer 2 Report
Report on paper"Anisotropy of the ΔE effect in Ni-based magnetoelectric cantilevers: a finite element method analysis"submitted by Hähnlein et al., for publication in Sensors (sensors-1764258).
The authors investigated the anisotropy of the ΔEeffect and its impact on the sensitivity of magnetoelectric sensors based on microelectromechanical cantilevers consisting of TiN / AlN / Ni. While the paper is interesting, it cannot be accepted in its present form and the authors must perform some modifications by addressing the following comments:
1- All the details about the Finite Element Model (Model size, type of elements, solver, convergence, stability) should be given and discussed in the paper.
2- In the modeling part, the used assumptions must be stated and justified for the governing equations.
3- Given the size of the studied device and the thickness of the layers, is it accurate to use the classical Euler Bernoulli model or a second strain gradient theory is more relevant to take into account possible nonlocal effects?
4- In the results, the authors should discuss in more details the single crystal approximation of the individual layers and its influence on the simulations.
5- What about the dynamic range of the studied sensor?
6- In the conclusion, the limitations of the proposed model should be specified from a critical point of view.
Author Response

(The authors gave the same response as above.)

Reviewer 3 Report
Tonisch and coworkers have reported a finite element method analysis of a Ni-based cantilever. The paper is well written and easy to follow. The authors have reported a study for the three principal magnetic directions of the Ni substrate used on top of the cantilever (i.e. easy, intermediate and hard direction) because the Young’s modulus is highly dependent on the direction of the field. Interestingly, Ni has a negative magnetostriction. The influence of the magnetostriction of the eigenfrequency of the cantilever was studied as well. I found particularly useful Fig 3a where the deflection of the tip is reported as a function of the Ni thickness. The results show that the resultant sensitivities are comparable to the ones reported for soft materials. While reading the paper I did not see any major flaw, therefore I am happy to recommend publication after minor revisions.
1) Figure 1 lacks the a and b panels.
2) Table 1 is difficult to understand: some derivatives are mentioned more than once with different sensitivities. Please clarify.
Author Response

(The authors gave the same response as above.)

Round 2
Reviewer 1 Report
Authors have improved this version of manuscript considering the comments of reviewer. This revised manuscript can be accepted for publication in Sensors.
Reviewer 2 Report
The authors have addressed my comments sufficiently to recommend publication of the paper in its current form.